# Message framing to inform cancer prevention pricing interventions in the UK and USA: a factorial experiment, 2019

Joseph G L Lee [ID] ,[1,2] Julie V Cristello [ID] ,[3] Christina H Buckton,[4] Rachel N Carey,[5] Elisa M Trucco,[6,7] Paulina M Schenk,[5] Theresa Ikegwuonu,[4] Shona Hilton [ID] ,[4] Shelley D Golden,[2,8] David I Conway[9]

For numbered affiliations see end of article.

**Correspondence to**
Dr Joseph G L Lee;
LEEJOSE14@ecu.edu

## ABSTRACT

**Objectives** To advance understanding of how message framing can be used to maximise public support across different pricing policies for alcohol, tobacco and sugary drinks/foods that prevent consumption of cancer-causing products.

**Design** We designed a 3×4×3 randomised factorial experiment to test responses to messages with three pricing policies, four message frames and three products.

**Setting** Online survey panel (Qualtrics) in 2019.

**Participants** Adults (N=1850) from the UK and USA.

**Interventions** Participants randomly viewed one of 36 separate messages that varied by pricing policy (increasing taxes, getting rid of price discounts, getting rid of low-cost products), four frames and product (alcohol, tobacco, sugary drinks/foods).

**Primary and secondary outcome measures** We assessed the relationship between the message characteristics and four dependent variables. Three were related to policy support: (1) increasing taxes on the product mentioned in the message, (2) getting rid of price discounts and special offers on the product mentioned in the message and (3) getting rid of low-cost versions of the product mentioned in the message. One was related to reactance, a psychological response to having one's freedom limited.

**Results** We found no effect for pricing policy in the message. Frames regarding children and reducing cancer risk moderated some outcomes, showing promise for real-world use. We found differences in support by product and reactance with greatest support and least reactance for tobacco policies, less support and more reactance for alcohol policies, and the least support and most reactance for sugary drinks/foods policies.

**Conclusions** Cancer prevention efforts using policy interventions can be informed by the message framing literature. Our results offer insights for cancer prevention advocacy efforts across the UK and USA and highlight that tax versus non-tax approaches to increasing the cost of cancer-causing products result in similar responses from consumers.

## Strengths and limitations of this study

► We conducted qualitative formative work and a pilot study to inform the experimental design and stimuli.
► Our team and data represented both the UK and the USA.
► We have strong internal validity given our use of a randomised factorial design.
► Our sample size was based on a power calculation for small effect sizes.
► Testing messages in a controlled experiment likely has limited ecological validity given real-world exposure to multiple messages across multiple channels.

## INTRODUCTION

One-third of the risk and burden of cancer is attributed to four common risk factors—use of tobacco, use of alcohol, unhealthy diets and physical inactivity.[1] These individual behaviours are influenced by a complex array of factors, including people's perceptions and beliefs as well as macrolevel drivers, such as corporate marketing of unhealthy products including product price.[2 3] Pricing interventions that increase the price of harmful products are an effective measure governments have to help encourage consumers to reduce the consumption of products that contribute to cancer.[4–12] There are multiple ways to increase costs of products. Many alcohol, sugary drinks/foods and tobacco products can be subject to excise taxes, which levy a specific or proportional fee on a product that is collected as revenue by the levying government. In addition to taxation, emerging research identifies non-tax price-raising strategies, including policies that set a minimum product price; mitigation fees that recoup public costs from product use; minimum pack sizes that prevent very cheap product prices; minimum excise taxes that ensure a

certain level of taxation; restricting sales to government stores; and, bans on product price discounts and other price marketing.[9 13–16] WHO considers such policies that raise the cost of products 'best buys' for preventing and controlling non-communicable diseases.[17]

Despite strong evidence for the effectiveness of price-related policies, the implementation of such policies has been limited, due in part to industry efforts to undermine public and political support.[18–20] Indeed, substantial portions of the public remain sceptical of tax interventions and they are often described as nanny state policies or government overreach.[21–25] Despite this scepticism, support for efforts to protect children and to give everyone a comparable opportunity of achieving a healthy life are broadly supported.[26] This disconnect between a lack of public support for price-related policy interventions, yet strong support for protecting people's health represents an opportunity to reframe the debate.

Public health advocates must translate potentially effective policies into ideas that resonate with public concerns.[27] The field of message framing, based in sociology and prospect theory from behavioural economics, suggests that how the information is presented changes how it is received and interpreted.[28 29] There is good evidence showing that the framing of policies in the media or in communication campaigns can play a critical role in the adoption of policies.[25 30–33] For example, in the USA, research has suggested that mixing individual and community responsibility frames can reduce counter arguments from more conservatively oriented members of the public, and linking other popular policies to a proposed policy can be an important strategy (eg, a tax is connected with financing universal early childhood education or a 'public health levy').[26 34 35] Some types of messages can perform well across the left-to-right ideological spectrum including focusing on military readiness, healthcare costs and reducing obesity-related bullying.[36] However, messages identifying social inequities may perform poorly by invoking discriminatory stereotypes from the more advantaged groups and negative emotions from more disadvantaged groups.[26 37 38] In Scotland, framing minimum unit alcohol pricing policy as a public health and whole-population issue has been crucial to enabling policymakers to adopt policies.[33] Some research also suggests that showing the effectiveness of a policy can have a small, positive effect on policy support.[39]

As such, framing can be an important tool in public health advocacy efforts to gain support for policies that address corporate determinants of health.[40 41] Prior research on framing has enumerated frames used to address health-harming products.[42 43] In addition to understanding the role of the policy and the message's framing, it is important to know if there are differences in support by the type of product. Differences in support by the product could inform advocacy campaigns—especially if advocates move from a product-by-product approach to an umbrella approach for cancer-causing products.

To this end, as part of a US National Cancer Institute and Cancer Research UK sponsored Knowledge Integration Sandpit,[44] we formed a binational, interdisciplinary team to explore the use of framing in cancer-prevention pricing policies. We hypothesised that (Hyp$_1$) messages about non-tax pricing policies would result in greater levels of support for pricing policies than messages about tax policies given a strong distaste for taxes as a policy intervention[23]; (Hyp$_2$) pricing policies would be differentially supported based on different frames devised through a qualitative development process and (Hyp$_3$) tobacco-related policies would be more supported than alcohol- or sugary drinks/foods-related policies.

## METHODS
### Message frame development and formative interviews
This factorial experimental design contained three stages and a sample drawn from the UK and US public. In stage 1, we conducted qualitative interviews with adults from the UK and adults from the USA between February and May 2018. In total, we held 18 interviews, nine in each country (three members of the public, three cancer policy advocates and three cancer survivors). We leveraged community partnerships and, in the USA, student researchers' connections to churches and other organisations. We purposively sampled for diversity by age, gender and race/ethnicity. Trained interviewers in Scotland and North Carolina conducted interviews in their respective countries using a semistructured interview guide. Briefly, we asked participants about their initial thoughts, advantages/disadvantages and benefits/harms of four pricing policies (minimum prices, getting rid of coupons or discounts, adding taxes and mitigation fees). We then asked about level of support for policies targeting specific products: alcohol, sugary drinks/foods and tobacco. Finally, we asked what arguments for a pricing policy would be persuasive. All interviews were professionally transcribed verbatim and transcriptions checked by the original interviewer. Transcripts were entered into QSR International's NVivo V.12 qualitative data analysis software for thematic analysis.[45] Two researchers read all transcripts in depth and carried out initial coding following the interview guide structure. We then inductively identified recurrent themes for each price policy and compared similarities and differences in views between price policies, products and countries.

In stage 2, we used the results from the formative qualitative interviews and the published literature[22 26 33 43] to develop nine potential frames and iteratively refined them for clarity and readability (a description of the potential frames is available in our repository[46]). For example, based on suggestions from the Robert Wood Johnson Foundation,[26] we developed a frame highlighting individual and community responsibility: 'We can all play a role in making healthier communities. Everyone can contribute to this by making healthy choices regarding their own alcohol consumption, but

**Table 1** Example frames used in experiment

| Frame no (short description) | Example frame for alcohol product and tax policy |
| --- | --- |
| $F_1$ (reduce strain on healthcare) | The problems caused by alcohol misuse, such as crime, injuries and addiction, are costly. Increasing taxes on alcohol will reduce the financial strain on our overburdened healthcare system. |
| $F_2$ (protect children) | Alcohol misuse can hurt children's health and well-being—both directly (through children drinking alcohol), and indirectly (through adults' misuse of alcohol impacting children). We can protect children by increasing taxes on alcohol. |
| $F_3$ (prevent cancer) | Drinking less alcohol can reduce the risk of cancer. Increasing taxes on alcohol is an important way to prevent cancer in our communities. |
| $F_4$ (bring consumption down) | Drinking less alcohol can reduce the risk of stroke, high blood pressure, obesity, liver disease and mental health problems. Increasing taxes on alcohol is an important way to bring drinking levels down. |

also by supporting increasing taxes on alcohol.' We then pilot-tested the frames with 106 UK and US adults in an online survey panel provided by Qualtrics Research Services, 30 October 2018–1 November 2018. Participants reported how well they understood the potential frames on a 3-point scale (1=understood very well, 2=understood fairly well, 3=did not understand well), described each in their own words and ranked the nine frames from most to least compelling. To select the frames for our main experiment, we computed the self-reported mean comprehensibility and the average ranking for each frame, and double coded (on a 3-point scale) whether participants accurately described the frame. In this step we did not manipulate product or policies; all messages were about alcohol taxes.

In the third and final stage, we selected four frames for an experiment. The four frames (table 1) were adapted to each of three products (alcohol, sugary drinks/foods, tobacco) and each of three policies (taxes, minimum pricing, getting rid of discounting). We created 36 messages specific to product, policy, and the four frames about how the policy intervention would (1) reduce strain to healthcare, (2) protect children, (3) prevent cancer and (4) bring consumption of the product down. All messages are provided in our linked Dataverse repository.

### Experimental design and implementation

We conducted an a priori power analysis for the 3×4×3 factorial design using G*Power V.3.1[47] with power (1 − β) set at 0.80 and α=0.05, two tailed, for a small effect size (Cohen's f=0.10) given prior findings examining similar constructs.[35 36] The power analysis indicated that a sample of 1745 would be needed to detect these small effects. To recruit participants, we engaged Qualtrics Research Services, an online panel service; this approach shows evidence of generalisability for experiments.[48 49] We used quota sampling to include 50% of participants from the UK and 50% from the USA, equal representation by assigned birth sex, a minimum of 20% of participants with fewer than 4 years of postsecondary education, and a minimum of 20% of participants reporting having smoked 100 or more cigarettes in one's life. Qualtrics provided each participant 'points' in exchange for completing

the survey that could be redeemed for incentives such as gift cards. Qualtrics successfully recruited and provided data from 1850 participants. In this between-subjects experimental factorial design, each participant was randomly assigned to view one of the 36 messages using Qualtrics's advanced block programming. Table 2 shows the experiment design with blue text for the cancer product and red text for the policy.

After brief screening and demographic questions, we provided the following prompt: 'On the next screen, we will show you a message about a public health policy and a product. After you read the message, we'll ask you questions about what you think about policies about that product. Please read the message about the product on the next screen. You will not be able to see it again.' The participant then viewed one of the 36 messages.

### Measures

To assess the impact of the different products, policies, and frames in our messages, we used four dependent variables. Following prior work by Niederdeppe *et al*,[35] we first measured policy support with three questions: 'How much do you oppose or support the following?'

**Table 2** Frame × products × policy factorial design

| | Policy | | |
| --- | --- | --- | --- |
| | **Tax** | **Getting Rid of Discounting** | **Min. Price** |
| Cancer product | | | |
| Alcohol | $F_1$ A T | $F_1$ A D | $F_1$ A P |
| | $F_2$ A T | $F_2$ A D | $F_2$ A P |
| | $F_3$ A T | $F_3$ A D | $F_3$ A P |
| | $F_4$ A T | $F_4$ A D | $F_4$ A P |
| Sugary drinks/ foods | $F_1$ S T | $F_1$ S D | $F_1$ S P |
| | $F_2$ S T | $F_2$ S D | $F_2$ S P |
| | $F_3$ S T | $F_3$ S D | $F_3$ S P |
| | $F_4$ S T | $F_4$ S D | $F_4$ S P |
| Tobacco | $F_1$ T T | $F_1$ T D | $F_1$ T P |
| | $F_2$ T T | $F_2$ T D | $F_2$ T P |
| | $F_3$ T T | $F_3$ T D | $F_3$ T P |
| | $F_4$ T T | $F_4$ T D | $F_4$ T P |

(1) Increasing taxes on the product mentioned in the message, (2) Getting rid of price discounts and special offers on the product mentioned in the message and (3) Getting rid of low-cost versions of the product mentioned in the message. There were five response options ranging from strongly oppose (1) to strongly support (5) with a neutral point. Each participant answered all three questions regardless of the policy invoked in their message frame. We also measured reactance, a psychological response to having one's freedom limited,[50] to the message using six items adapted from Hall *et al*,[51] for example, 'The product mentioned is legal, so the government should stop interfering with it.' There were five response options ranging from strongly disagree (1) to strongly agree (5) with a neutral point, which we averaged into a scale. These items exhibited good internal consistency in our data (McDonald's omega=0.87) and could not be improved by removing any item.[52]

We also measured potential covariates and factors including country (UK, USA), and use of alcohol, sugary beverages and cigarettes in the last 30 days (options: every day, some days, not at all). Full details of measures are available in our repository.

### Analysis of experiment

We assessed whether the: (1) type of policy, (2) product or (3) frame used in a message was associated with support for each type of policy and reactance, as well as interactions among these main predictors. We also assessed interactions between covariates (sex, age, country) and main predictors; if they were significant, those covariates were analysed as an additional factor as was the case for country in two of the models. All main effects and interactions were analysed using a 3×4×3 analysis of variance (ANOVA). In two of the models, where country was significant, a 2×3×4×3 ANOVA was conducted. We chose ANOVA over a regression framework as an ANOVA approach provides equivalent results to linear regression with dummy-coded predictors[53] and we did not wish to compare results to a single reference category. When main effects or interactions were significant, we conducted pairwise comparisons using Fisher's least significant differences test. We used a traditional threshold of statistical significance, $p < 0.05$, and used two tailed tests. We used SPSS V.25 (IBM).

Verbal consent was obtained from all interviewees and online survey experiment participants received study information and were asked to continue to the study if they agreed to participate.

### Patient and public involvement

Patients and the public were not involved in this research.

# RESULTS
## Formative interviews

We completed formative interviews with eighteen participants, nine in the UK and nine in the USA. Interviewees were between ages 24 and 69 years (average age 48), and cancer survivors had a range of cancer experiences including prostate, lung, breast, head and neck, and skin cancers. Interviews lasted between twenty minutes and 1 hour depending on the interviewees' interest in the subject area. In general, interviewees were more supportive of price policies for tobacco containing commodities than either alcohol or sugary drinks/foods (table 3).

During the interviews, it became apparent that participants found it difficult to differentiate between a hypothecated tax and a mitigation fee. The team discussed this issue and decided to drop mitigation fee as a policy in the main experiment. A number of emergent themes were common to all three pricing policies (table 4). Specifically, participants thought that all of the policies would have an impact on purchasing behaviour with a concomitant health benefit due to reduced consumption, particularly for young people and those on low income. Participants also thought they could have an indirect benefit by drawing attention to the product and the level of potential health harm. For example, a policy advocate and cancer survivor said, respectively:

'…it would reduce the excessive consumption of these products, it would control the quantities in which these products are consumed, and it would mean the consumer… is more aware of their purchasing of the products.' (Policy advocate)

The biggest benefit is that it will make the person aware—some people drink and eat things they're not aware of, they don't look into it—and if they see that there's a higher tax, and they start recognising that, they're going to find out why and it might change their behaviour.' (Cancer survivor)

**Table 3** Interviewees' overall support for price policies in general by product type, country and interviewee type, N=18

| Product | UK interviewees | | | | | | | | | US interviewees | | | | | | | | | Mean |
|---|---|---|---|---|---|---|---|---|---|---|---|---|---|---|---|---|---|---|---|
| | Advocates | | | Survivors | | | Public | | | Advocates | | | Survivors | | | Public | | | |
| Tobacco | 9 | 10 | 8 | 8 | 9 | 10 | 1 | 9 | 9 | 8 | 9 | 8 | 8 | 5 | 8 | 7 | 8 | 8 | 7.9 |
| Alcohol | 9 | 10 | 4 | 3 | 7 | 7 | 1 | 5 | 6 | 7 | 6 | 4 | 3 | 5 | 5 | 4 | 8 | 6 | 5.6 |
| Sugary drinks/foods | 10 | 10 | 3 | 8 | 4 | 8 | 3 | 4 | 9 | 5 | 3 | 2 | 7 | 5 | 2 | 5 | 6 | 4 | 5.4 |

1=not supportive, 10=very supportive.

**Table 4** Themes identified in qualitative interviews

| Common for all three pricing policies | Specific to minimum price | Specific to getting rid of discounts | Specific to tax |
|---|---|---|---|
| Supportive arguments/advantages | | | |
| ► Health benefit<br>► Greatest benefit for young<br>► Greatest benefit for low SES<br>► Will affect purchasing behaviour<br>► Will highlight real cost of consumption/health harms<br>► More supportive where there is clear evidence of cancer links | ► A deterrent or nudge<br>► Consistency and fairness<br>► Targets most harmful consumption<br>► There is evidence to support it | ► Price promotions most strongly influence purchasing<br>► Promotions encourage over consumption<br>► Will protect industry and retailer profits | ► Could prompt reformulation<br>► Could be a form of health insurance<br>► Source of public revenue to fund social policy programmes/ health services<br>► An effective way of moderating behaviour |
| Unsupportive arguments/disadvantages | | | |
| ► Effectiveness will depend on individual response<br>► Needs to be prohibitively high to change consumption<br>► Unfair on responsible consumers<br>► Addicts will pay more<br>► Could create a black market (US specific)<br>► Regressive<br>► Bad for retailers/producers<br>► Simplistic—not enough | ► Contravenes EU law<br>► Public do not understand/will not support it<br>► Revenue raised goes to industry<br>► Too specific for food<br>► Will not impact population level drinking | ► Lost opportunity to apply to healthy foods<br>► Will push prices up across the board<br>► Inappropriate to interfere in pricing | ► Nanny statist—interference in freedom of choice<br>► Political cost—unpopular<br>► Reformulation could prompt increased consumption<br>► Negative unintended consequences—substitutes may be equally unhealthy<br>► Will not work if funds are not ring fenced |

EU, European Union; SES, socioeconomic status.

Some key disadvantages were also highlighted across all policies: (1) that their effectiveness would depend on the scale of the price increase and how individuals responded; (2) that they are potentially regressive and, (3) that they are unfair to responsible consumers and to those who are dependent on these substances. Participants were especially concerned about the impact on individuals dependent on these products with limited resources, particularly in relation to alcohol consumption. Yet, this concern also applied to tobacco and sugary drinks/foods, which were both deemed to have addictive qualities. Quotes illustrating these disadvantages include:

The effect on health does depend on what decisions people make in response to it. And whether it follows what we expect. And does it reduce purchase of those products, does it push people to other products or not? (Policy advocate)

I think they [people with limited incomes] would then be more likely to stop buying food, to stop heating their house, or to stop paying their rent or, you know, whatever, in order to prioritize the alcohol consumption.' (General public)

There were a number of unique themes arising for each price policy (table 3). For example, although discounts and price promotions were thought to be the strongest influencer of purchasing behaviour and to encourage over consumption, interviewees thought they should be refocused on healthier foods and drinks rather than replaced. For minimum unit price policies specifically,

interviewees noted that an advantage of such policies is that they target the most harmful patterns of consumption, that is, the highest levels of consumption which can result in antisocial behaviour. Taxation alone prompted interviewees to talk about nanny state arguments and inappropriate government intervention in the market. This response was more marked in the UK than the USA and might be explained by the fact that the timing of the interviews coincided with the implementation of the UK Soft Drinks Industry Levy ('Sugar Tax') in April 2018.

The frames that interviewees thought would be most persuasive and result in increased support for price policies could be summarised as those that: (1) draw a very clear link to potential health harm from consuming the product; (2) focus on a positive health impact arising from the policy, particularly protection of children; (3) emphasise the impact on health services in terms of reduced costs and the potential to generate revenue for reinvestment and (4) make the policies meaningful in terms of the anticipated benefit for individuals in straight forward language. Less convincing frames were those that: (1) employ threats or scare tactics; (2) seem condescending or nanny statist; (3) use jargon or statistics and (4) set unrealistic expectations of behavioural change for individuals. We used these findings to inform the experimental stimuli, results for which are presented next.

### Main experimental effects
Participants were 1850 adults (ages 18–86, $M_{age}$=38.56, SD=13.56). The sample was evenly split across the countries

**Table 5** Summary of significance direction of main effects and interactions between factors and dependent variables, 2019, N=1850, UK and US adults

| Factors | Dependent variables | | | |
| | Support for raising taxes | Support for getting rid of price discounts and special offers | Support for getting rid of low-cost version of product | Psychological reactance |
| --- | --- | --- | --- | --- |
| Policy (taxes, minimum pricing, getting rid of discounting) | ▶ Main: NS<br>▶ Interactions: NS | ▶ Main: NS<br>▶ Interactions: NS | ▶ Main: NS<br>▶ Interactions: NS | ▶ Main: NS<br>▶ Interactions: NS |
| Frame (1. reduce strain to healthcare, 2. protect children, 3. prevent cancer and 4. bring consumption of the product down) | ▶ Main: NS<br>▶ Interactions: NS | ▶ Main: Higher support with frame 2 (Children) and 3 (Prevent cancer) than 4 (Reduce Use)<br>▶ Interaction: NS | ▶ Main: NS<br>▶ Interaction: For participants receiving frame 3 (prevent cancer) support was higher for minimum pricing policy in message than for tax policy in message<br>▶ For participants receiving frame 4 (Reduce use) differences in support by policy in message | ▶ Main: NS<br>▶ Interaction: frame 2 (Children) lowered reactance for UK participants more than US participants |
| Product (alcohol, sugary drink/food, tobacco) | ▶ Main: Tobacco: Highest; Alcohol: Lower; Sugar: Lowest;<br>▶ Interaction: Tobacco higher for UK than USA | ▶ Main: Tobacco: Highest; Alcohol: Lower; Sugar: Lowest<br>▶ Interaction: NS | ▶ Main: Tobacco: Highest; Alcohol: Lower; Sugar: Lowest<br>▶ Interaction: NS | ▶ Main: Sugar: Highest; Alcohol: Lower; Tobacco: Lowest<br>▶ Interaction: Sugar and Tobacco lower for UK than USA |
| Country (UK, USA)* | ▶ Main: UK: Higher; USA: Lower<br>▶ Interaction: NS | ▶ Main: NS<br>▶ Interaction: NS | ▶ Main: NS<br>▶ Interaction: NS | ▶ Main: USA: Higher; UK: Lower<br>▶ Interaction: NS |

Sugar=sugary drinks/foods, NS=not statistically significant.
*Covariate assessed as a factor given significant interactions with other factors; interactions assessed were between covariates (sex, age, country) and main predictors (policy, frame, product; if they were significant, those covariates were analysed as an additional factor as was the case for country in two of the models.

of interest (50.1% from the UK) and sex assigned at birth (50.2% female). Within the sample, 34.7% graduated from college or university, 26.4% attended some college or university, 28.9% had graduated or completed secondary or high school and the remainder (7.6%) attended some high school or less. Table 5 shows the pattern of results.

### Dependent variable 1: support for increasing taxes on the product mentioned in the message

For the first outcome, there was not a significant main effect for policy (ie, taxes, minimum pricing, getting rid of discounting) or frame (ie, reduce strain to healthcare, protect children, prevent cancer and bring consumption of the product down). There was a significant main effect for product ($F_{(2, 1767)}=45.71$, $p<0.001$) and country ($F_{(1, 1767)}=26.63$, $p<0.001$). Pairwise comparisons for product revealed significant differences across all three products ($M_{sugar}=2.76$, SD=1.26; $M_{alcohol}=3.07$, SD=1.30; $M_{tobacco}=3.47$; SD=1.37); participants receiving a message regarding sugary drinks/foods had the lowest and participants receiving a message regarding tobacco endorsed the highest levels of support for increasing taxes on the product. Pairwise comparisons for country revealed that participants from the UK ($M_{UK}=3.26$, SD=1.33) endorsed

higher levels of support for increasing taxes compared with participants from the US ($M_{US}=2.94$, SD=1.34). There was also a significant two-way interaction: product × country, $F_{(2, 1767)}=5.67$, $p<0.01$. Specifically, among participants receiving a sugary drinks/foods product message, those from the UK endorsed higher levels of support for increasing taxes on the product ($M_{UK}=3.08$, SD=1.28) compared with US participants ($M_{US}=2.51$, SD=1.19). Similarly, among participants receiving a tobacco product message, those from the UK endorsed higher levels of support for increasing taxes on the product ($M_{UK}=3.69$, SD=1.32) compared with US participants ($M_{US}=3.24$, SD=1.38). Three-way interactions were not significant.

### Dependent variable 2: support for getting rid of price discounts and special offers on the product mentioned in the message

For the second outcome, there was not a significant main effect for policy (ie, taxes, minimum pricing, getting rid of discounting). There was a significant main effect for frame ($F_{(3, 1803)}=2.62$, $p<0.05$) and message product ($F_{(2, 1803)}=53.80$, $p<0.001$). Participants receiving frames 2 (protect children, $M_{frame2}=3.25$, SD=1.37) or 3 (prevent cancer, $M_{frame3}=3.24$, SD=1.35) endorsed higher

levels of support for getting rid of price discounts and special offers than participants who received frame 4 (bring consumption down, $M_{frame4}$=3.04, SD=1.36). Pairwise comparisons for product revealed significant differences across all three products ($M_{sugar=}$2.85, SD=1.28; $M_{alcohol}$=3.05, SD=1.34; $M_{tobacco}$=3.59; SD=1.35); participants receiving a message regarding sugary drinks/foods and participants receiving a message regarding tobacco endorsed the lowest and highest levels of support for getting rid of price discounts and special offers on the product, respectively. There were no significant two-way or three-way interactions.

### Dependent variable 3: support for getting rid of low-cost versions of the product mentioned in the message

For the third outcome, there was not a significant main effect for policy (ie, taxes, minimum pricing, getting rid of discounting) or frame (ie, reduce strain to healthcare, protect children, prevent cancer and bring consumption of the product down). There was a significant main effect for message product (F(2, 1802)=60.34, p<0.001). Pairwise comparisons revealed significant differences across all three products ($M_{sugar}$=2.82, SD=1.26; $M_{alcohol}$=3.08, SD=1.29; $M_{tobacco}$=3.60; SD=1.34), such that participants receiving a message regarding sugary drinks/foods and participants receiving a message regarding tobacco endorsed the lowest and highest levels of support for getting rid of low-cost versions of the product, respectively. There was also a significant two-way interaction: frame × policy, F(6, 1802)=2.24, p<0.05. Specifically, among participants receiving frame 3 (prevent cancer), participants who received the policy related to minimum pricing endorsed higher levels of support for getting rid of low-cost versions of the product ($M_{minprice}$=3.36, SD=1.28) compared with those receiving the policy related to increasing taxes ($M_{tax}$=3.05, SD=1.38). Additionally, among participants receiving frame 4 (bring consumption down), there were significant differences in support for getting rid of low-cost products between participants who received policies related to getting rid of discounting ($M_{nodiscounting}$=3.23, SD=1.30) and minimum pricing ($M_{minprice}$=3.16, SD=1.37), and participants who received the policy related to increasing taxes ($M_{tax}$=2.76, SD=1.29). There were no significant three-way interactions.

### Dependent variable 4: reactance

For the last outcome, there was not a significant main effect for policy or frame on reactance (ie, the extent to which one perceives their freedoms to be limited). There was a significant main effect for message product (F(2, 1769)=30.74, p<0.001) and country (F(1, 1769)=7.21, p<0.01). Pairwise comparisons for product revealed significant differences across all three products ($M_{tobacco}$=2.72; SD=0.99; $M_{alcohol}$=2.92, SD=0.95; $M_{sugar}$=3.14, SD=0.93), such that participants receiving a message regarding tobacco and participants receiving a message regarding sugary drinks/foods, indicated the lowest and highest levels of reactance, respectively. Pairwise comparisons

for country revealed that participants from the USA ($M_{US}$=2.99, *SD*=0.97) endorsed higher levels of reactance compared with participants from the UK ($M_{UK}$=2.87, SD=0.97). There were two significant two-way interactions: product × country, F(2, 1769)=4.09, p<0.05 and frame × country F(3, 1769)=2.84, p<0.05. Specifically, among those that received a message regarding tobacco, participants from the US endorsed significantly higher levels of reactance ($M_{US}$=2.84, SD=0.98) compared with participants from the UK ($M_{UK}$=2.61, SD=0.98). Similarly, among those receiving a message regarding sugary drinks/foods, participants from the US endorsed significantly higher levels of reactance ($M_{US}$=3.24, SD=0.91) compared with participants from the UK ($M_{UK}$=3.05, SD=0.93). Among those who received frame 2 (protect children), participants from the UK endorsed lower levels of reactance ($M_{UK}$=2.71, SD=0.97) than those from the US ($M_{US}$=3.01, SD=0.93).

Lastly, there was also a significant three-way interaction: product × policy × country, (F(4, 1769)=2.97, p<0.05). Among those receiving a message regarding alcohol, those from the UK endorsed significantly higher levels of reactance ($M_{UK}$=3.05, SD=0.92) compared with participants from the US ($M_{US}$=2.78, SD=0.94) receiving the getting rid of discounting policy. In contrast, among those receiving a message regarding sugary drinks/foods, those from the US endorsed significantly higher levels of reactance ($M_{US}$=3.39, SD=0.92) compared with participants from the UK ($M_{UK}$=3.07, SD=0.95) for those receiving the policy related to increasing taxes. Among those receiving a message regarding tobacco, those from the US endorsed significantly higher levels of reactance ($M_{US}$=2.93, SD=1.05) compared with participants from the UK ($M_{UK}$=2.48, SD=0.95) receiving the getting rid of discounting policy.

## DISCUSSION
### Principal findings

This research aimed to explore whether the specific content of a message about cancer-prevention price policies was associated with higher public support and lower reactance for those policies. We had three main hypotheses. First, we hypothesised that messages about non-tax pricing policies would result in more support than messages about tax policies. Our results did not provide evidence for this hypothesis: Not one of the three policies tested (taxes, minimum pricing, and getting rid of discounts) showed a statistically significant impact on support in our experiment. Regarding our second hypothesis, that support for a particular pricing policy would change based on the framing of the message, we found some evidence. Specifically, of the frames we tested, which were selected based on their performance in pilot testing, our findings indicate that frames relating to children show promise. We found evidence to support our third hypothesis that messages about tobacco products would result in more support for raising the price of

products than messages about alcohol or sugary drinks/foods products. Product in the message also changed reactance, which was highest for sugary drinks/foods and lowest for tobacco. Finally, we found statistically significant differences in responses to messages by country (eg, support for alcohol pricing interventions was lower in the USA than in the UK).

## Strengths and weaknesses

The strong internal validity from the experimental design, qualitative formative work and international scope of this research must be balanced against limitations. First, this study used an online convenience sample from a panel provider service, Qualtrics, which may limit its external validity given the participants are not representative of their respective countries, particularly in tobacco use and educational attainment. However, prior research indicates that convenience samples like ours generalise well when used in experimental studies rather than studies of prevalence.[48] Second, we assessed the impact of a single exposure of a message displayed on a screen. This does not reflect the real-world environment where messages about policies may be delivered from multiple sources, repeated and seen on multiple communication channels. Additionally, a limitation to survey research is that participants can interpret response scales differently which can impact data and interpretations. Thus, participants were provided with a wider range of responses (eg, 5-point Likert scales) and category labels to minimise these variations. Third, further work should explore equity implications of our messaging approaches—especially given growing evidence regarding the proequity effect of pricing interventions.[54]

## Results in context

The process of policy development as well as policy making decisions are ultimately taken by politicians on behalf of their constituents (the public); in this context, the scientific evidence and value bases play important roles.[55] While the evidence base for different types and targets of price policies on behaviour is rapidly expanding,[56–61] the difficulty of increasing public support and consequently political support for pricing interventions has received less attention. The relatively low levels of public support have previously been highlighted.[22 62] Our findings provide further evidence to suggest limited support among the public for pricing interventions overall, given relatively low endorsements of support on our support dependent variables. However, given the mean scores, the average participant was also not that opposed. Furthermore, a policy message that specifically referenced a particular type of price policy did not result in more support for that type of price policy compared with other types.

Prior research has examined the role of framing in public health efforts, and our findings demonstrate ways to best use media advocacy approaches[63] by minimising reactance from the public and promoting support of evidence-based policies. Specifically, our frames were all

selected based on the prior literature and their performance in formative work. Particular value may come from framing messages around the protection of children. This is consistent with previous work.[26] Previous work has also shown that the product under consideration influences policy support.[64] Our experimental findings offer further evidence of this, showing that participants rated support of pricing interventions highest on messages for tobacco products, followed by alcohol and lastly by sugary drinks/foods. Finally, our findings highlight some notable differences by country, which may reflect more general differences in acceptance of policies, healthcare systems and political developments at the time of our study (eg, 'Brexit').

## Implications for public health policy and practice

Our findings have several implications for messaging about pricing policies and for policy research, particularly with the aim of garnering support (across stakeholder groups from public, civil society, media and politicians) to develop and implement price policies. First, the evidence of differences by product may indicate that efforts to 'denormalise' tobacco products may have yet to translate to alcohol and sugary drinks/foods efforts. Policy advocates should consider if lessons learnt from tobacco control can and should be applied to alcohol and sugary drinks/foods. It is worth highlighting that denormalisation can present the risk of increasing stigma, which has been a criticism in the area of tobacco.[65] Second, our findings suggest the importance of bridging the research-practice gap between the literature on framing and advocacy organisations' messaging. Frames that include a focus on protecting children seem to show particular promise; however, our four frames showed few significant difference in their overall performance. This is likely because we used a robust pilot testing process to develop the frames, and we only selected the best performing frames for use in the main experiment. Practitioners should consider use of the four frames tested here. Third, practitioners should consider lessons learnt from other countries, which may help with identifying particular challenges and successes regarding cancer prevention.

## CONCLUSIONS

In our online experiment with participants from the UK and USA, we found no significant differences in price policy message support or reactance based on the type of proposed policy, suggesting advocates can recommend policies that raise prices through both tax and non-tax means. Results also indicate that framing messages focused on protecting children had increased support and lowered reactance in some combinations of experimental stimuli. Public health policy advocates should consider using frames like those we tested in ongoing efforts to develop, adopt and implement price policies that reduce use of cancer-causing products. This study also indicates that lessons learnt from efforts to build public

support for addressing tobacco use may be needed to successfully address alcohol and sugary drinks/products, since support for pricing policies for these products were generally lower than for tobacco price policies. Public health practitioners and advocates should consider using tested messages when working to advance pricing policies that reduce consumption of harmful products marketed by powerful corporations.

**Author affiliations**
[1]Department of Health Education and Promotion, College of Health and Human Performance, East Carolina University, Greenville, North Carolina, USA
[2]Cancer Prevention and Control, University of North Carolina Lineberger Comprehensive Cancer Center, Chapel Hill, North Carolina, USA
[3]Department of Psychology, Florida International University, Miami, Florida, USA
[4]MRC/CSO Social and Public Health Sciences Unit, Institute of Health & Wellbeing, University of Glasgow, Glasgow, UK
[5]Department of Clinical, Educational & Health Psychology, University College London, London, UK
[6]Department of Psychology and Center for Children and Families, Florida International University, Miami, Florida, USA
[7]Department of Psychiatry, University of Michigan, Ann Arbor, Michigan, USA
[8]Department of Health Behavior, Gillings School of Global Public Health, University of North Carolina at Chapel Hill, Chapel Hill, North Carolina, USA
[9]School of Medicine, Dentistry, and Nursing, College of Medical Veterinary and Life Sciences, University of Glasgow, Glasgow, UK

**Acknowledgements** We thank Cancer Research UK and the US National Cancer Institute Health Behaviours Research Branch for the opportunity to participate in their jointly hosted 2017 'sandpit' workshop. We thank Israel M. Mendez for his help with the data collection and project management in the USA. An earlier version of this paper was presented in a dissemination event held at the Royal Institution in London in November 2019.

**Contributors** JGLL, RNC, ET and DC conceived the study. JGLL, CHB, RNC, ET, TI, SH, SDG and DC designed the study. JVC (quantitative) and CHB (qualitative) conducted the data analysis with input from ET and SH, respectively. PMS conducted analyses of the pilot study with input from RNC. JGLL, CHB, RNC, ET, PMS, SDG and DC developed the stimuli. JGLL led the manuscipt drafting; all authors provided critical input on the interpretation of the results, writing of the manuscript and approved the final version.

**Funding** This work was supported by Cancer Research UK: BUPA Foundation Fund - International Innovation Grant (C18486/A25644), internal research funds at East Carolina University, UK Medical Research Council (MC_UU_12017/13, MC_UU_12017/15), the Chief Scientist Office of the Scottish Government Health Directorates (SPHSU13, SPHSU15) and the US National Institutes of Health (K08AA023290, T32DA043449).

**Disclaimer** The content is solely the responsibility of the authors and does not necessarily represent the official views of the funders.

**Competing interests** None declared.

**Patient and public involvement** Patients and/or the public were not involved in the design, or conduct, or reporting, or dissemination plans of this research.

**Patient consent for publication** Not required.

**Ethics approval** Research undertaken in this project was reviewed by the East Carolina University and Medical Center Institutional Review Board (#17–002904) and University of Glasgow College of Medical, Veterinary & Life Sciences Ethics Committee (#200170074).

**Provenance and peer review** Not commissioned; externally peer reviewed.

**Data availability statement** Data are available in a public, open access repository. Data, codebooks and interview guides are available in the East Carolina University Dataverse, doi:10.15139/S3/IAKUQN, available from: https://dataverse.unc.edu/dataset.xhtml?persistentId=doi:10.15139/S3/IAKUQN.

**ORCID iDs**
Joseph G L Lee http://orcid.org/0000-0001-9698-649X
Julie V Cristello http://orcid.org/0000-0002-7513-2206
Shona Hilton http://orcid.org/0000-0003-0633-8152

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
