## [Reviewer comments · BMJ Open]

ARTICLE DETAILS

TITLE (PROVISIONAL)	Message framing to inform cancer prevention pricing interventions in the UK and US: A factorial experiment, 2019
AUTHORS	Lee, Joseph; Cristello, Julie; Buckton, Christina; Carey, Rachel; Trucco, Elisa; Schenk, Paulina; Ikegwuonu, Theresa; Hilton, Shona; Golden, SD; Conway, D

VERSION 1 – REVIEW

REVIEWER	Christina Zorbas Deakin University, Australia
REVIEW RETURNED	03-Jul-2020

GENERAL COMMENTS	Thank you for the opportunity to review this manuscript outlining a study of public support and reactance to pricing policies and message frames across alcohol, tobacco and sugary products. Whilst this study addresses a very topical and important area of research, and uses a comprehensive approach to do so, some parts of the manuscript (particularly the results and discussion) could be articulated in a clearer manner. Overall: • When referring to ‘sugar’ products – additionally clarity is needed to highlight that you are not actually talking about bags of sugar, but sugar containing products (E.g. sugary drinks, etc.). It is unclear throughout, which of these products you investigated through your message framing. A similar issue arises when you refer to ‘sugar policies’ – are these taxes on sugary drinks or foods?• Consistent language could also be used to refer to ‘bans/getting rid/elimination of price discounts’ (i.e. line 64 of the discussion). Abstract: • Line 67: consider removing ‘used six items’ or explaining further as it is unclear what this related to. Introduction: • Lines 109-111: suggest replacing “common behaviours” with “common risk factors” and “poor dietary choices” with “unhealthy diets” or “diet risks” as this better aligns with public policy messaging (i.e. does not blame individuals for their behaviours).• Lines 110-112: when discussing the ‘complex array of factors’ that drive individual health outcomes, price should be discussed as a key driver.• Line 131: the term ‘framing’ needs to be defined.• Line 132-133: additional nuance could be added to the sentence ‘enumerated frames used to address health-harming products’. E.g. do you mean that ‘Prior research has identified frames that help promote efforts (by whom?) to address health-harming products’. Specific examples of this evidence may also be helpful. • Hypothesis 1 could be better supported by evidence in the introduction section to explicate why the authors held this
--

	hypothesis. This is particularly important as you did not find evidence of this (lines 401-402). As a whole, little evidence is presented for the effectiveness and prioritisation of each pricing policy investigated. Methods:  • Line 146: Please add a subheading to the first section of the methods (e.g. Frame development/Formative interviews) • Lines 150-151: Additional information on how participants were recruited for interviews could be provided, especially the members of the public and cancer survivors. Additional information on the 'sociodemographic diversity' could also be provided (sex, age?) Results:  • Line 165: when stating "to develop nine potential frames", it is unclear what these are frames of? The sentence seems incomplete. • Lines 294-296: It is unclear why reducing anti-social behaviour is a disadvantage of a minimum price policy? • Lines 305-206: It is also unclear what it means to 'make the policies real and relevant for people' • Table 4: A more descriptive title is required, as are more refined subtitles for the Dependent variables. As it is, this table does not stand alone, especially for the policy frame results where it is hard to tell which policy options are being compared. • Line 323: could the term "message product" be replaced with "product"? The former is somewhat ambiguous. • Lines 322-323/339-340/352-353/369-370: Each introductory line of the results could be written in a standalone manner where more context is provided and the reader does not need to refer back to the methods to understand. E.g. 'Policy or frame type did not have a significant main effect on policy reactance (i.e. the extent to which one perceives their psychological response to be limited).' • Lines 359-366: I do not understand the rationale for testing the relationship of policy support within policy type. This was not a hypothesis being tested as far as I understand. • Lines 368-396: It would be useful for readers if the definition of reactance is reiterated and if the meaning of a higher agreeance is incorporated throughout the results. Whilst interesting, the results are articulated in a way that is difficult to follow. E.g. Line 385: "the UK endorsed weaker agreement" which meant that they perceived that policy impinged on rights to a lesser extent than the US. Discussion:  • Lines 406-408: "Specifically, our findings indicate that frames relating to children and those relating to reducing risk of cancer may provide the most promise." I am not sure that comparative statements like this are warranted when the results simply compare whether there are significant differences regarding the support of policy frames and do not quantify the magnitude of these differences. Thus, how can you conclude which frame is provides the most promise? You can only infer that one frame might provide more promise than another. Any discussion of the actual effectiveness of the frames appears to be limited by their inadequate presentation of these findings in the results. • Lines 416-418: In what ways is the external validity limited? Additional consideration of the equity implications of this research should also be indicated, especially given the overrepresentation of highly educated groups and how pricing policies hold promise for equitably improving population health. • Additional discussion of the rich framing literature is required (i.e. how do the results of this study compare to what has already been found?). • The implications of this study for public health policy and practice
--	--

	could be more specifically addressed (possibly under a subheading). Conclusions  • The conclusion needs to be considerably refined so that new ideas aren't introduced (e.g. 'it is worth highlighting the denormalization can present the risk of increasing stigma...' – this is not mentioned anywhere else in the manuscript). The conclusions should concisely come back to the overall aims, implications and significance of this study.
--	---

REVIEWER	Michelle Scollo Cancer Council Victoria
REVIEW RETURNED	16-Aug-2020

GENERAL COMMENTS	This was a cleverly designed study examining an important area of public policy. And the report is very well written. I feel, however, that attempting to cover three different products in two different countries limited the usefulness of the findings. It is already well known that there is greater support for policies addressing tobacco than there is for policies addressing alcohol and sugary drinks. It is already known that the public support for policies to restrict corporate promotion of unhealthy products tends to be lower in the US than in other English-speaking countries. It is also well known that the public is more supportive of policies if it is understood how these would protect children or provide strong benefits to public health such as preventing cancer. The finding about the lack of difference in support for tax versus non-tax policies was interesting. This suggests that the research participants were not overly worried about the framing of the intervention as a 'tax'. If the authors are keen to see their work published in a high profile journal, then perhaps this aspect of the findings could be highlighted in a much shorter paper or letter.
---

REVIEWER	Rob Branston University of Bath, UK
REVIEW RETURNED	24-Aug-2020

GENERAL COMMENTS	This is an interesting and important paper on the question of how to 'frame' market interventions for tobacco, alcohol and sugar, in order to get the maximum public support for policies to reduce consumption. The paper is timely, and is generally of a suitable quality for publication. That said there are a number of issues, that if addressed, would improve the quality of the paper. I only really have two issues of significance, with a number of other minor points of detail that should be easily accommodated. Firstly, the paper does not at any point qualify what it means by 'frames'. This is a major weakness as the idea of framing is a key concept of the paper so this needs to be clearly done in order allow the reader to fully access the ideas inherent in the paper. This strikes me as a marketing type of terms but it certainly isn't one that I am familiar with. My second and biggest concern are the statistical methods
---

employed. To me the use of ANOVA is overly simplistic and it isn't clear to me why this approach has been utilised when I think the results might have been better explored using OLS estimation of some specified equations for the relationships being explored. Perhaps that is just my economic background talking, but I think the approach chosen needs better justification in the methodology, and then also discussed within the strengths/weaknesses section (which does seem very brief). At present it neither discussed the data used or the method chosen which I would have expected to see.

Specific Points

P.3, line 20/21 – what is “four frames” – I don't understand.

P.5, line 15 “price per unit” – I'm not sure the ‘per unit’ is helpful here as it is suggestive of minimum unit pricing. Perhaps simply “increase the price of harmful products” is sufficient at this point

P.5 – lines 26 to 36. What about other policies that are seen in the area of tobacco such as minimum pack sizes that increase pack purchase prices, and bans on price marking of packets (to keep retail prices low)? You might also mention things like sale restriction, that only allow alcohol to be sold in government stores (thereby preventing price discounting). You might even mention tax related interventions such as Minimum Excise Taxes which look to modify regular tax systems in order to address the cheapest products in a market. I appreciate you list can't be exhaustive but I think it can offer a bit more to give a wider sense of the options available.

P.6 – line 6 “... to gain public and policy support for policies...” In this context ‘policy’ is inappropriate. Do you mean ‘policy maker’, ‘politicians’ or something like that perhaps? Policy per se can't give support.

P.6 – lines 6-8: “Prior research 133 on framing has enumerated frames used to address health-harming products” - I think this needs to be explained in more detail. What exactly do you mean by the use of ‘frames’.

P.7 – lines 24 to 34. The 3 hypotheses should be better separated to aid the reader gain a clear understanding. At the very least they needed to be separated by semi-colons. Furthermore, see the comment above about ‘frames’, which means as written I don't understand Hyp2.

P.7 – methodology – you need to be clear on what you mean by ‘frames’

P.7, line 53. You want 20% of the sample to have been former smokers. Did you ensure similar numbers were regular (or former) drinkers, and also users of sugar? Either way why have this 20% threshold? Smoking rates are not that high in either country so I'm wondering if this might add bias. At the very least this 20% for current or former tobacco users needs to be justified.

P10/11 – why did you use ANOVA rather than simply creating some specified equations and then running these using OLS (or similar) regression? This would have allowed you to simultaneously consider all variables and the differences that you want to explore? I'm not saying that you had to adopt this approach, I would just like to see more justification that the method used was the best, most

	appropriate tool to use. Using such an equation/OLS approach would make the results easier to understand and offered far more powerful results in my opinion. P.20 – you have no discussion of the limitations of your method. This to me is very surprising indeed. You need to discuss both your method and your data. For instance, you are talking about a number of 5 point likert scales for your results (although you don't name it as such) and seem to convert these into a number. While I have no issue with this as an approach, some do, and I think you have to acknowledge the issues in doing this – for instance, individuals have different interpretations of the scale and the difference between a 4 and a 5 might not be the same as between a 3 and a 4.
--	--

VERSION 1 – AUTHOR RESPONSE

Reviewer: 1

Reviewer Name: Christina Zorbas

Institution and Country: Deakin University, Australia

Please state any competing interests or state 'None declared': None declared

Please leave your comments for the authors below

Thank you for the opportunity to review this manuscript outlining a study of public support and reactance to pricing policies and message frames across alcohol, tobacco and sugary products. Whilst this study addresses a very topical and important area of research, and uses a comprehensive approach to do so, some parts of the manuscript (particularly the results and discussion) could be articulated in a clearer manner.

AUTHOR RESPONSE: We appreciate your constructive review. We have worked to address the comments below and in combination with those of the other reviewers.

Overall:

- When referring to 'sugar' products – additionally clarity is needed to highlight that you are not actually talking about bags of sugar, but sugar containing products (E.g. sugary drinks, etc.). It is unclear throughout, which of these products you investigated through your message framing. A similar issue arises when you refer to 'sugar policies' – are these taxes on sugary drinks or foods?

AUTHOR RESPONSE: We have made this consistent and clarified our language. Specifically, we now say "sugary drinks/foods" throughout.

- Consistent language could also be used to refer to 'bans/getting rid/elimination of price discounts' (i.e. line 64 of the discussion).

AUTHOR RESPONSE: We have made this change. We now consistently use “getting rid of” given this parallels the actual language used in the stimuli.

Abstract:

- Line 67: consider removing ‘used six items’ or explaining further as it is unclear what this related to.

AUTHOR RESPONSE: We have removed this language.

Introduction:

- Lines 109-111: suggest replacing “common behaviours” with “common risk factors” and “poor dietary choices” with “unhealthy diets” or “diet risks” as this better aligns with public policy messaging (i.e. does not blame individuals for their behaviours).

AUTHOR RESPONSE: We have made this change.

- Lines 110-112: when discussing the ‘complex array of factors’ that drive individual health outcomes, price should be discussed as a key driver.

AUTHOR RESPONSE: We have made this change. Specifically, it now reads, “These individual behaviours are influenced by a complex array of factors, including people’s perceptions, beliefs as well as macro-level drivers such as corporate marketing of unhealthy products including product price.”

- Line 131: the term ‘framing’ needs to be defined.

AUTHOR RESPONSE: We’ve better described and defined the concept of framing. We now note and describe their psychological/behavioral economics and sociological origins. We also provide examples of how framing has been used in public health advocacy efforts. Specifically, we added “Public health advocates must translate potentially effective policies into ideas that resonate with public concerns.²⁵ The field of message framing, based in sociology and prospect theory from behavioural economics, suggests that how the information is presented changes how it is received and interpreted.^{26,27} There is good evidence showing that the framing of policies in the media or in communication campaigns can play a critical role in the adoption of policies.^{23,28-31} For example, in the US, research has suggested that mixing individual and community responsibility frames can reduce counter arguments from more conservatively-oriented members of the public, and linking other popular policies to a proposed policy can be an important strategy (e.g., a tax is connected with financing universal early childhood education or a “public health levy”).^{24,32,33} Some types of messages can perform well across the left-to-right ideological spectrum including focusing on military

readiness, healthcare costs, and reducing obesity-related bullying.³⁴ However, messages identifying social inequities may perform poorly by invoking discriminatory stereotypes from the more advantaged groups and negative emotions from more disadvantaged groups.^{24,35,36} In Scotland, framing minimum unit alcohol pricing policy as a public health and whole-population issue has been crucial to enabling policymakers to adopt policies.³¹ Some research also suggests that showing the perceived effectiveness of a policy can have a small, positive effect on policy support.³⁷

- Line 132-133: additional nuance could be added to the sentence 'enumerated frames used to address health-harming products'. E.g. do you mean that 'Prior research has identified frames that help promote efforts (by whom?) to address health-harming products'. Specific examples of this evidence may also be helpful.

AUTHOR RESPONSE: We think we have addressed this comment by providing examples of framing in the preceding paragraph, which addresses the reviewer's comment to ensure there is appropriate nuance in how we are describing the literature. The sentence itself is correct as written, as the citations are content analyses identifying frames used but not testing them. Thus, we have not changed this sentence.

- Hypothesis 1 could be better supported by evidence in the introduction section to explicate why the authors held this hypothesis. This is particularly important as you did not find evidence of this (lines 401-402). As a whole, little evidence is presented for the effectiveness and prioritisation of each pricing policy investigated.

AUTHOR RESPONSE: We appreciate the reviewer's comment. We are trying to maintain a reasonable word count, and thus we have not further described the existing literature on evidence for how each mechanism for increasing prices is effective. Rather, and we hope this is acceptable to the reviewer, our introduction makes the case that changing the price influences behavior. (Specifically, it notes: "Pricing interventions that increase the price per unit of harmful products are an effective measure governments have to help encourage consumers to reduce the consumption of products that contribute to cancer."⁴⁻¹² There are multiple ways to increase per-unit costs of products. Many alcohol, sugary drink/food, and tobacco products can be subject to excise taxes, which levy a specific or proportional fee on a product that is collected as revenue by the levying government. In addition to taxation, emerging research identifies *non-tax* price-raising strategies, including policies that set a minimum product price, mitigation fees that recoup public costs from product use, like litter collection, and get rid of product price discounts.^{9,13,14} The World Health Organization considers such policies "best buys" for preventing and controlling non-communicable diseases.¹⁵)

We hope this conveys that the legal mechanism through which the price is increased may not matter if the consumer is faced with a higher price. That said, we did ensure that we are citing evidence of the different pricing intervention approaches in the introduction. The reviewer's point on the directionality of the hypothesis is also well taken. We have added additional information to show it is based on a particular dislike of taxes. (Put simply, the idea behind the directionality of this hypothesis is that simply saying the word "tax" may reduce the effectiveness of the message.) Specifically, it now reads, "We hypothesized that (**Hyp₁**) messages about non-tax pricing policies would result in greater levels of support for pricing policies than messages about tax policies given a strong distaste for taxes

as a policy intervention,²¹.” We cite: Petrescu, D. C., Hollands, G. J., Couturier, D. L., Ng, Y. L., & Marteau, T. M. (2016). Public Acceptability in the UK and USA of Nudging to Reduce Obesity: The Example of Reducing Sugar-Sweetened Beverages Consumption. *PLoS One*, 11(6), e0155995. doi:10.1371/journal.pone.0155995

Methods:

- Line 146: Please add a subheading to the first section of the methods (e.g. Frame development/Formative interviews)

AUTHOR RESPONSE: We have made this change by adding a subheading: “Message frame development and formative interviews”

- Lines 150-151: Additional information on how participants were recruited for interviews could be provided, especially the members of the public and cancer survivors. Additional information on the ‘sociodemographic diversity’ could also be provided (sex, age?)

AUTHOR RESPONSE: We note that we leveraged networks from community partnerships and, in the USA, our student researchers’ connections to churches and other organizations. Specifically, it now reads, “We leveraged community partnerships and, in the US, student researchers’ connections to churches and other organizations. We purposively sampled for diversity by age, gender, and race/ethnicity.”

Results:

- Line 165: when stating “to develop nine potential frames”, it is unclear what these are frames of? The sentence seems incomplete.

AUTHOR RESPONSE: We have clarified our language, noting that our process for selecting frames resulted in nine potential frames, and providing an example. Specifically, it now reads, “In stage two, we used the results from the formative qualitative interviews and the published literature^{20,24,31,41} to develop nine potential frames and iteratively refined them for clarity and readability (a description of the potential frames is available in our repository⁴⁴). For example, based on suggestions from the Robert Wood Johnson Foundation²⁴ we developed a frame highlighting individual and community responsibility: ‘We can all play a role in making healthier communities. Everyone can contribute to this by making healthy choices regarding their own alcohol consumption, but also by supporting increasing taxes on alcohol.’”

- Lines 294-296: It is unclear why reducing anti-social behaviour is a disadvantage of a minimum price policy?

AUTHOR RESPONSE: In Table 3 we indicate that one of the supportive arguments/advantages given for minimum unit pricing was that it targets the most harmful consumption. We have clarified this in the text which now reads 'For minimum unit price policies specifically, interviewees noted that an advantage of such policies is that they target the most harmful patterns of consumption.'

- Lines 305-206: It is also unclear what it means to 'make the policies real and relevant for people'

AUTHOR RESPONSE: We have clarified our meaning and the statement now reads '(4) make the policies meaningful in terms of the anticipated benefit for individuals, in straight forward language.'

- Table 4: A more descriptive title is required, as are more refined subtitles for the Dependent variables. As it is, this table does not stand alone, especially for the policy frame results where it is hard to tell which policy options are being compared.

AUTHOR RESPONSE: We have revised the title and subtitles of Table 4. The title is now: "Summary of significance and interactions between factors and dependent variables, 2019, N=1,850, UK and US adults". We have modified the subtitle of each row to better describe what factors are being summarized. What previously read Frame, for example, now reads: "Frame (1. reduce strain to healthcare, 2. protect children, 3. prevent cancer, and 4. bring consumption of the product down)."

- Line 323: could the term "message product" be replaced with "product"? The former is somewhat ambiguous.

AUTHOR RESPONSE: We have made this change.

- Lines 322-323/339-340/352-353/369-370: Each introductory line of the results could be written in a standalone manner where more context is provided and the reader does not need to refer back to the methods to understand. E.g. 'Policy or frame type did not have a significant main effect on policy reactance (i.e. the extent to which one perceives their psychological response to be limited).'

AUTHOR RESPONSE: We have made these changes. For example, for the first outcome, which the reviewer indicates above as line 322-323 in the original submission, read "For the first outcome, there was not a significant main effect for policy or frame." Per the reviewer's suggestion, we have revised it to read: "For the first outcome, there was not a significant main effect for policy (i.e., taxes, minimum pricing, getting rid of discounting) or frame (i.e., reduce strain to healthcare, protect children, prevent cancer, and bring consumption of the product down)." We have made parallel edits for the other outcomes including the reviewer's suggested edit related to reactance. We have also relabeled the subheadings from "Outcomes" to "Dependent variables" to parallel the table and methods.

- Lines 359-366: I do not understand the rationale for testing the relationship of policy support within policy type. This was not a hypothesis being tested as far as I understand.

AUTHOR RESPONSE: We apologize for any confusion. To clarify: policy support was an outcome or dependent variable. There were four outcomes or dependent variables examined overall, three of which related to policy support (for three different policies). Specifically, these outcomes were: 1) policy support for increasing taxes on the product mentioned in the message, 2) policy support for getting rid of price discounts and special offers on the product mentioned in the message, and 3) policy support for getting rid of low cost versions of the product mentioned in the message. The scale for policy support, across these three outcomes, was 1 = “*strongly oppose*” to 5 = “*strongly support*.” The fourth outcome was reactance, based on a subset of items from the Modified Reactance to Health Warning Scale. The scale for these items was 1 = “*strongly disagree*” to 5 = “*strongly agree*.” Additional information about the models is detailed on pages 10-11.

- Lines 368-396: It would be useful for readers if the definition of reactance is reiterated and if the meaning of a higher agreement is incorporated throughout the results. Whilst interesting, the results are articulated in a way that is difficult to follow. E.g. Line 385: “the UK endorsed weaker agreement” which meant that they perceived that policy impinged on rights to a lesser extent than the US.

AUTHOR RESPONSE: We have made these changes. First, we added a repeated definition of reactance at the start of this section (it now reads: “For the last outcome, there was not a significant main effect for policy or frame on reactance (i.e., the extent to which one perceives their freedoms to be limited).”). Additionally, we edited this section to remove agreement and disagreement with the scale and specifically talk about lower versus higher reactance. For example, the line 385 result in the original submission now reads, “Among those who received frame 2 (protect children), participants from the UK endorsed lower levels of reactance ($M_{UK}=2.71$, $SD=0.97$) than those from the US ($M_{US}=3.01$, $SD=0.93$).”

Discussion:

- Lines 406-408: “Specifically, our findings indicate that frames relating to children and those relating to reducing risk of cancer may provide the most promise.” I am not sure that comparative statements like this are warranted when the results simply compare whether there are significant differences regarding the support of policy frames and do not quantify the magnitude of these differences. Thus, how can you conclude which frame is provides the most promise? You can only infer that one frame might provide more promise than another. Any discussion of the actual effectiveness of the frames appears to be limited by their inadequate presentation of these findings in the results.

AUTHOR RESPONSE: We have revised this and it now reads: “Specifically, of the frames we tested, which were selected based on their performance in pilot testing, our findings indicate that frames relating to children show promise.”

- Lines 416-418: In what ways is the external validity limited? Additional consideration of the equity implications of this research should also be indicated, especially given the overrepresentation of highly educated groups and how pricing policies hold promise for equitably improving population health.

AUTHOR RESPONSE: We have made these changes. Specifically, the limitations now read, “First, this study used an online convenience sample from a panel provider service, Qualtrics, which may limit its external validity given the participants are not representative of their respective countries, particularly in tobacco use and educational attainment. However, prior research indicates that convenience samples like ours generalize well when used in experimental studies rather than studies of prevalence.” AND, regarding equity, “Third, further work should explore equity implications of our messaging approaches – especially given growing evidence regarding the pro-equity effect of pricing interventions.⁵¹”

- Additional discussion of the rich framing literature is required (i.e. how do the results of this study compare to what has already been found?).

AUTHOR RESPONSE: As part of our response to comments above, we added an extensive discussion of framing, with relevant examples, to the introduction. Given the high word count, we do not want to repeat that here, and we think the revised paper gives enough context for a reader without further expanding our word count. The discussion reads: “Prior research has examined the role of framing in public health efforts, and our findings demonstrate ways to best utilize media advocacy approaches⁶⁰ by minimizing reactance from the public and promoting support of evidence-based policies. Specifically, our frames were all selected based on the prior literature and their performance in formative work. Particular value may come from framing messages around the protection of children and reduction of cancer risk. Previous work has shown that the product under consideration influences policy support.⁶¹ Our experimental findings offer further evidence of this, showing that participants rated support of pricing interventions highest on messages for tobacco products, followed by alcohol, and lastly by sugary drinks/foods.” We hope this is acceptable to the reviewer.

- The implications of this study for public health policy and practice could be more specifically addressed (possibly under a subheading).

AUTHOR RESPONSE: We have edited the conclusion per the next comment and separated out implications for policy and practice, which includes a new subheading. Specifically, it now reads, **“Implications for Public Health Policy and Practice**

Our findings have several implications for messaging about pricing policies and for policy research, particularly with the aim of garnering support (across stakeholder groups from public, civic society to media and politicians) to develop and implement price policies. First, the evidence of differences by product may indicate that efforts to ‘denormalize’ tobacco products may have yet to translate to alcohol and sugary drink/food efforts. Policy advocates should consider if lessons learned from tobacco control can and should be applied to alcohol and sugary drinks/foods. It is worth highlighting that denormalization can present the risk of increasing stigma, which has been a criticism in the area of tobacco.⁶² Second, our findings suggest the importance of bridging the research-practice gap

between the literature on framing and advocacy organizations' messaging. Frames that include a focus on protecting children seem to show particular promise; however, our four frames did not show significant difference in their overall performance. This is likely because we used a robust pilot testing process to develop the frames, and we only selected the best performing frames for use in the main experiment. Practitioners should consider use of the four frames tested here. Third, practitioners should consider lessons learned from other countries, which may help with identifying particular challenges and successes regarding cancer prevention.”

Conclusions

- The conclusion needs to be considerably refined so that new ideas aren't introduced (e.g. 'it is worth highlighting the denormalization can present the risk of increasing stigma...' – this is not mentioned anywhere else in the manuscript). The conclusions should concisely come back to the overall aims, implications and significance of this study.

AUTHOR RESPONSE: We have revised our conclusion to remove any new ideas and streamline it with a proper connection back, as the reviewer suggests. The conclusion now reads: “In our online experiment with participants from the UK and US, we found no significant differences in price policy message support or reactance based on the type of proposed policy, suggesting advocates can recommend policies that raise prices through both tax and non-tax means. Results also indicate that framing messages focused on protecting children had increased support and lowered reactance in some combinations of experimental stimuli. Public health policy advocates should consider using frames like those we tested in ongoing efforts to develop, adopt, and implement price policies that reduce use of cancer-causing products. This study also indicates that lessons learned from efforts to build public support for addressing tobacco use may be needed to successfully address alcohol and sugary drinks/products, since support for pricing policies for these products were generally lower than for tobacco price policies. Public health practitioners and advocates should consider using tested messages when working to advance pricing policies that reduce consumption of harmful products marketed by powerful corporations.”

Reviewer: 2

Reviewer Name: Michelle Scollo

Institution and Country: Cancer Council Victoria

Please state any competing interests or state 'None declared': None declared

Please leave your comments for the authors below

This was a cleverly designed study examining an important area of public policy. And the report is very well written.

I feel, however, that attempting to cover three different products in two different countries limited the usefulness of the findings.

It is already well known that there is greater support for policies addressing tobacco than there is for policies addressing alcohol and sugary drinks.

It is already known that the public support for policies to restrict corporate promotion of unhealthy products tends to be lower in the US than in other English-speaking countries.

It is also well known that the public is more supportive of policies if it is understood how these would protect children or provide strong benefits to public health such as preventing cancer.

The finding about the lack of difference in support for tax versus non-tax policies was interesting. This suggests that the research participants were not overly worried about the framing of the intervention as a 'tax'. If the authors are keen to see their work published in a high profile journal, then perhaps this aspect of the findings could be highlighted in a much shorter paper or letter.

AUTHOR RESPONSE: We appreciate your review and appreciate the kind comment regarding the design of the study. We agree that some of our findings are consistent with the existing literature. We have opted to keep them in a combined manuscript in line with our originally proposed hypotheses, and because we think the findings are most useful to researchers and practitioners when considered in the context of the study as a whole. Much of the value in this research, we think, comes from the fact that it builds on individual previous research findings, across different disciplines and countries, and draws them together in a new study.

Reviewer: 3

Reviewer Name: Rob Branston

Institution and Country: University of Bath, UK

Please state any competing interests or state 'None declared': None declared

Please leave your comments for the authors below

This is an interesting and important paper on the question of how to 'frame' market interventions for tobacco, alcohol and sugar, in order to get the maximum public support for policies to reduce consumption. The paper is timely, and is generally of a suitable quality for publication. That said there are a number of issues, that if addressed, would improve the quality of the paper.

AUTHOR RESPONSE: Thank you for your comments, which have improved the reporting of our research.

I only really have two issues of significance, with a number of other minor points of detail that should be easily accommodated. Firstly, the paper does not at any point qualify what it means by 'frames'. This is a major weakness as the idea of framing is a key concept of the paper so this needs to be clearly done in order allow the reader to fully access the ideas inherent in the paper. This strikes me as a marketing type of terms but it certainly isn't one that I am familiar with.

AUTHOR RESPONSE: We've better described and defined the concept of framing. We now note their psychological/behavioral economics and sociological origins and describe them along with providing examples. Specifically, we added "Public health advocates must translate potentially effective policies into ideas that resonate with public concerns."²⁵ The field of message framing, based in sociology and prospect theory from behavioural economics, suggests that how the information is presented changes how it is received and interpreted.^{26,27} There is good evidence showing that the framing of policies in the media or in communication campaigns can play a critical role in the adoption of policies.^{23,28-31} For example, in the US, research has suggested that mixing individual and community responsibility frames can reduce counter arguments from more conservatively-oriented members of the public, and linking other popular policies to a proposed policy can be an important strategy (e.g., a tax is connected with financing universal early childhood education or a "public health levy").^{24,32,33} Some types of messages can perform well across the left-to-right ideological spectrum including focusing on military readiness, healthcare costs, and reducing obesity-related bullying.³⁴ However, messages identifying social inequities may perform poorly by invoking discriminatory stereotypes from the more advantaged groups and negative emotions from more disadvantaged groups.^{24,35,36} In Scotland, framing minimum unit alcohol pricing policy as a public health and whole-population issue has been crucial to enabling policymakers to adopt policies.³¹ Some research also suggests that showing the perceived effectiveness of a policy can have a small, positive effect on policy support.³⁷

My second and biggest concern are the statistical methods employed. To me the use of ANOVA is overly simplistic and it isn't clear to me why this approach has been utilised when I think the results might have been better explored using OLS estimation of some specified equations for the relationships being explored. Perhaps that is just my economic background talking, but I think the approach chosen needs better justification in the methodology, and then also discussed within the strengths/weaknesses section (which does seem very brief). At present it neither discussed the data used or the method chosen which I would have expected to see.

AUTHOR RESPONSE: We appreciate the reviewer's thoughtfulness around the statistical methods used in this study. While the reviewer suggests that one method may be superior, a regression with dummy-coded predictors would provide equivalent results to an ANOVA with between subject factors (Cohen et al., 2003). Additionally, an ANOVA is typically used to examine differences between groups while regressions examine relationships and make predictions. Thus, we have not made this change to our analysis.

Specific Points

P.3, line 20/21 – what is "four frames" – I don't understand.

AUTHOR RESPONSE: We've better described and defined the concept of framing. We now note their psychological/behavioral economics and sociological origins and describe them, as noted above.

P.5, line 15 "price per unit" – I'm not sure the 'per unit' is helpful here as it is suggestive of minimum unit pricing. Perhaps simply "increase the price of harmful products" is sufficient at this point

AUTHOR RESPONSE: We have made this change.

P.5 – lines 26 to 36. What about other policies that are seen in the area of tobacco such as minimum pack sizes that increase pack purchase prices, and bans on price marking of packets (to keep retail prices low)? You might also mention things like sale restriction, that only allow alcohol to be sold in government stores (thereby preventing price discounting). You might even mention tax related interventions such as Minimum Excise Taxes which look to modify regular tax systems in order to address the cheapest products in a market. I appreciate you list can't be exhaustive but I think it can offer a bit more to give a wider sense of the options available.

AUTHOR RESPONSE: Thank you. We have revised the list to read: "In addition to taxation, emerging research identifies *non-tax* price-raising strategies, including policies that set a minimum product price; mitigation fees that recoup public costs from product use; minimum pack sizes that prevent very cheap product prices; minimum excise taxes that ensure a certain level of taxation; restricting sales to government stores; and bans on product price discounts and other price marketing.^{9,13-16} The World Health Organization considers such policies that raise the cost of products "best buys" for preventing and controlling non-communicable diseases.¹⁷"

P.6 – line 6 "... to gain public and policy support for policies..." In this context 'policy' is inappropriate. Do you mean 'policy maker', 'politicians' or something like that perhaps? Policy per se can't give support.

AUTHOR RESPONSE: We have made this change by removing all adjectives from support.

P.6 – lines 6-8: "Prior research 133 on framing has enumerated frames used to address health-harming products" - I think this needs to be explained in more detail. What exactly do you mean by the use of 'frames'.

AUTHOR RESPONSE: We've better described and defined the concept of framing. We now note their psychological/behavioral economics and sociological origins and describe them. Specifically, we added "Public health advocates must translate potentially effective policies into ideas that resonate with public concerns.²⁵ The field of message framing, based in sociology and prospect theory from behavioural economics, suggests that how the information is presented changes how it is received and interpreted.^{26,27} There is good evidence showing that the framing of policies in the media or in

communication campaigns can play a critical role in the adoption of policies.^{23,28-31} For example, in the US, research has suggested that mixing individual and community responsibility frames can reduce counter arguments from more conservatively-oriented members of the public, and linking other popular policies to a proposed policy can be an important strategy (e.g., a tax is connected with financing universal early childhood education or a “public health levy”).^{24,32,33} Some types of messages can perform well across the left-to-right ideological spectrum including focusing on military readiness, healthcare costs, and reducing obesity-related bullying.³⁴ However, messages identifying social inequities may perform poorly by invoking discriminatory stereotypes from the more advantaged groups and negative emotions from more disadvantaged groups.^{24,35,36} In Scotland, framing minimum unit alcohol pricing policy as a public health and whole-population issue has been crucial to enabling policymakers to adopt policies.³¹ Some research also suggests that showing the perceived effectiveness of a policy can have a small, positive effect on policy support.^{37”}

P.7 – lines 24 to 34. The 3 hypotheses should be better separated to aid the reader gain a clear understanding. At the very least they needed to be separated by semi-colons. Furthermore, see the comment above about ‘frames’, which means as written I don’t understand Hyp2.

AUTHOR RESPONSE: We have made this change by adding semi-colons. As noted above, we’ve better described frames, which we believe addresses the reviewer’s second point.

P.7 – methodology – you need to be clear on what you mean by ‘frames’

AUTHOR RESPONSE: We’ve better described and defined the concept of framing as noted above, which we believe addresses the reviewer’s comment.

P.7, line 53. You want 20% of the sample to have been former smokers. Did you ensure similar numbers were regular (or former) drinkers, and also users of sugar? Either way why have this 20% threshold? Smoking rates are not that high in either country so I’m wondering if this might add bias. At the very least this 20% for current or former tobacco users needs to be justified.

AUTHOR RESPONSE: We considered the consumption of alcohol and sugary products to be common and wanted to ensure that if we controlled for product use, we would have enough smokers in our study. You are correct that this may limit the generalizability of our results, and we have added a note to the limitations to reflect this. Specifically, the limitations now read, “First, this study used an online convenience sample from a panel provider service, Qualtrics, which may limit its external validity given the participants are not representative of their respective countries, particularly in tobacco use and educational attainment. However, prior research indicates that convenience samples like ours generalize well when used in experimental studies rather than studies of prevalence.”

P10/11 – why did you use ANOVA rather than simply creating some specified equations and then running these using OLS (or similar) regression? This would have allowed you to simultaneously consider all variables and the differences that you want to explore? I’m not saying that you had to

adopt this approach, I would just like to see more justification that the method used was the best, most appropriate tool to use. Using such an equation/OLS approach would make the results easier to understand and offered far more powerful results in my opinion.

AUTHOR RESPONSE: We appreciate the reviewer’s thoughtfulness around the statistical methods used in this study. While the reviewer suggests that one method may be superior, a regression with dummy-coded predictors would provide equivalent results to an ANOVA with between subject factors (Cohen et al., 2003). Additionally, an ANOVA is typically used to examine differences between groups while regressions examine relationships and make predictions. Thus, we have not made a change to our analysis.

P.20 – you have no discussion of the limitations of your method. This to me is very surprising indeed. You need to discuss both your method and your data. For instance, you are talking about a number of 5 point likert scales for your results (although you don’t name it as such) and seem to convert these into a number. While I have no issue with this as an approach, some do, and I think you have to acknowledge the issues in doing this – for instance, individuals have different interpretations of the scale and the difference between a 4 and a 5 might not be the same as between a 3 and a 4.

AUTHOR RESPONSE: We thank the reviewer for their comment, as this is a limitation that is often not recognized in survey research. Varying interpretations of response scales, as well as extreme response styles, and midpoint response styles can alter the data and results of research studies. Prior work has found that increasing the number of response categories leads to higher reliability (Krosnick, 1991). Similarly, adding categories to 5-point rating scales has also been found to increase reliability (Preston & Coleman, 2000; Weng, 2004). Thus, we have attempted to minimize these concerns by providing participants with a wider range of responses (e.g., 5-point Likert scales), and category labels (e.g., strongly support vs. support). We now include this as a limitation on page 21.

VERSION 2 – REVIEW

REVIEWER	Christina Zorbas Deakin University, Australia
REVIEW RETURNED	24-Nov-2020
GENERAL COMMENTS	Thank you for the opportunity to review a revised version of this manuscript describing an experimental study of responses to 3 pricing policies, 4 message frames and 3 products among a sample of participants from the UK and USA. The authors have improved the quality of their manuscript and it will make an important contribution to the literature on how child protection and other frames may be used to effectively shift public health policy rhetoric. My final comment is that in Line 352 I believe Table 5 should be cited in text rather than Table 5.

REVIEWER	Rob Branston University of Bath, UK
REVIEW RETURNED	26-Nov-2020

GENERAL COMMENTS	I once again enjoyed reading this paper as I found it to be both interesting and important. I thought the authors had generally done a good job at responding to all of the queries raised in the reviews, and generally I'm pretty satisfied it is ready for publication. However, I have one remaining minor issue in terms of the statistical technical adopted that I would like to see addressed prior to publication. I am happy for the authors to continue with the ANOVA approach if that is what they prefer (although I don't agree with their characterising regression techniques as being about predictions - they are about assessing relationships between variables) but do think it needs to be explicitly justified as being appropriate within the text. This should only be a matter of citing a few references in the text and with a sentence or two of wider justification.
---

VERSION 2 – AUTHOR RESPONSE

Reviewer: 1

Reviewer Name: Christina Zorbas

Institution and Country: Deakin University, Australia

Please state any competing interests or state 'None declared': None declared

Comments to the Author

Thank you for the opportunity to review a revised version of this manuscript describing an experimental study of responses to 3 pricing policies, 4 message frames and 3 products among a sample of participants from the UK and USA. The authors have improved the quality of their manuscript and it will make an important contribution to the literature on how child protection and other frames may be used to effectively shift public health policy rhetoric.

My final comment is that in Line 352 I believe Table 5 should be cited in text rather than Table 4.

AUTHOR RESPONSES: Thank you for your review and for catching that typo, which we have fixed.

Reviewer: 3

Reviewer Name: Rob Branston

Institution and Country: University of Bath, UK

Please state any competing interests or state 'None declared': None known

Comments to the Author

I once again enjoyed reading this paper as I found it to be both interesting and important. I thought the authors had generally done a good job at responding to all of the queries raised in the reviews, and generally I'm pretty satisfied it is ready for publication. However, I have one remaining minor issue in terms of the statistical technical adopted that I would like to see addressed prior to publication.

I am happy for the authors to continue with the ANOVA approach if that is what they prefer (although I don't agree with their characterising regression techniques as being about predictions - they are about assessing relationships between variables) but do think it needs to be explicitly justified as being appropriate within the text. This should only be a matter of citing a few references in the text and with a sentence or two of wider justification.

AUTHOR RESPONSE: Thank you for your review of our paper and kind words. We have made this change. Specifically, we added the following rationale to the methods: "We chose ANOVA over a regression framework as an ANOVA approach provides equivalent results to linear regression with dummy-coded predictors[CITATION] and we did not wish to compare results to a single reference category" (lines 260-262). The citation was for: Cohen, J., Cohen, P., West, S. G., & Aiken, L. S. (2003). *Applied multiple regression/correlation analysis for the behavioral sciences* (3rd ed.). Lawrence Erlbaum Associates Publishers.